# Vitamin D Receptor Signaling Regulates Craniofacial Cartilage Development in Zebrafish

**DOI:** 10.3390/jdb7020013

**Published:** 2019-06-22

**Authors:** Hye-Joo Kwon

**Affiliations:** Biology Department, Princess Nourah University, Riyadh 11671, Saudi Arabia; xhjkwon@hotmail.com or hjkwon@pnu.edu.sa

**Keywords:** Vitamin D receptor (VDR), craniofacial cartilage, zebrafish, *follistatin a* (*fsta*)

## Abstract

Vitamin D plays essential roles in supporting the skeletal system. The active form of vitamin D functions through the vitamin D receptor (VDR). A hereditary vitamin-D-resistant rickets with facial dysmorphism has been reported, but the involvement of VDR signaling during early stages of craniofacial development remains to be elucidated. The present study investigated whether VDR signaling is implicated in zebrafish craniofacial cartilage development using a morpholino-based knockdown approach. Two paralogous *VDR* genes, *vdra* and *vdrb*, have been found in zebrafish embryos. Loss-of-*vdra* has no discernible effect on cartilage elements, whereas loss-of-*vdrb* causes reduction and malformation of craniofacial cartilages. Disrupting both *vdra* and *vdrb* leads to more severe defects or complete loss of cartilage. Notably, knockdown of *vdrb* results in elevated expression of *follistatin a* (*fsta*), a bone morphogenetic protein (BMP) antagonist, in the adjacent pharyngeal endoderm. Taken together, these findings strongly indicate that VDR signaling is required for early craniofacial cartilage development in zebrafish.

## 1. Introduction

Vitamin D is known to be involved in mineral homeostasis and the maintenance of a healthy skeleton throughout life [1]. Vitamin D deficiency is the primary cause of congenital rickets [2]. In children with vitamin-D-resistant rickets, craniofacial malformations are often observed [3]. The biological actions of vitamin D can be mediated by vitamin D receptor (VDR)-dependent and -independent pathways [4]. Loss-of-function mutations in the *VDR* gene cause hereditary vitamin-D-resistant rickets (HVDRR) in infancy [5]. Arita et al. [6] reported a familial HVDRR case with facial dysmorphism associated with homozygosity for a missense mutation in the *VDR* gene. However, it is not clear whether the *VDR* mutation itself leads to the craniofacial defects in this case. Several mouse models with targeted ablation of the VDR had been generated [7,8,9,10], which displayed different results; in one study, VDR-deficient mice had dysmorphism including a flat face and a short nose [10], but in another study, no facial abnormality was exhibited [8]. Although *VDR* transcript is nearly ubiquitous [11,12], relatively strong expression of VDR was found in the cartilage chondrocytes of neonatal mouse calvaria, the upper part of the neurocranium [13] and in the developing mandible in 48 hours post-fertilization (hpf) zebrafish embryo [14]. Thus, the role of the VDR signaling in craniofacial development remains unresolved and requires further exploration.

The formation of craniofacial structures is a complex process involving multiple tissue interactions [15,16]. Despite morphological differences, the general patterning and genetic network of craniofacial development are highly conserved across vertebrates [17,18]. The zebrafish system has been utilized as a valuable animal model for craniofacial research and provided much of our current understanding of craniofacial development [19,20,21]. In zebrafish, the craniofacial skeleton can be observed as early as 3 days post-fertilization (dpf), which is composed of cartilages later replaced by bone [22,23]. The craniofacial cartilages are mainly derived from cranial neural crest cells migrating into the pharyngeal arches and their proper differentiation is regulated by a combination of intrinsic cues and extrinsic signals from surrounding tissues [24,25]. The formation of craniofacial structures is controlled by multiple signaling molecules including fibroblast growth factor (FGF), sonic hedgehog (SHH), WNT, Endothelin-1 (Edn1), Notch, and bone morphogenetic protein (BMP) [26,27,28,29,30]. BMP signaling has long been recognized as a key regulator of the craniofacial development [31,32]. BMP ligands are expressed in developing pharyngeal arches [33,34]. The pharyngeal endoderm also expresses several BMP antagonists for fine-tuning of BMP activity [35,36,37]. Perturbation of BMP pathway results in craniofacial defects in mammals and zebrafish [38,39,40]. Furthermore, BMP signaling interacts with other signaling pathways to regulate dorsal-ventral patterning of the craniofacial skeleton [38,41,42].

The present study reports that VDR signaling is required for craniofacial cartilage formation in zebrafish embryo and downregulates expression of *follistatin a* (*fsta*), a BMP antagonist, in the adjacent pharyngeal endoderm for the proper craniofacial cartilage development.

## 2. Materials and Methods

### 2.1. Zebrafish Strain, Development, and Staging

The wild-type strain was derived from the AB line (Eugene, OR, USA). Embryos were maintained in an incubator at 28.5 °C, and staged as described [43]. All protocols for experiments complied with the US National Research Council’s Guide for the Care and IACUC-approved Animals Use Protocol number 2012–011.

### 2.2. Morpholino Oligomer Injection

Gene knockdown experiments using antisense morpholino oligomers (MOs) were carried out as previously described [44,45]. Morpholino oligomers (MOs) were obtained from Gene Tools, LLC (Philomath, OR, USA). and diluted in Danieau solution (58 mM NaCl, 0.7mM KCl, 0.4 mM MgSO_4_, 0.6 mM Ca(NO_3_)_2_, 5.0 mM HEPES, pH 7.6). Zebrafish embryos were injected with 5 ng of each MOs at the one-cell stage. All MOs used here have been validated and reported [44,46,47]. To knockdown *vdra*, a translation blocker (5′-AACGGCACTATTTTCCGTAAGCATC-3′) was used. To knockdown *vdrb*, a splice blocker (5′-TCCATCACTAGCAGACGAGGGAAGA-3′), which targets the intron2-exon3 (I2E3) junction was used. In all *vdrb* knockdown experiments, embryos were coinjected with *p53* MO to ensure inhibition of nonspecific cell death as described [48]. To assess phenotypes, at least 10 embryos were examined in each experiment. The phenotypes described in this study were completely penetrant. The experiments were conducted at least three times.

### 2.3. Alcian Blue Staining and Whole-Mount In Situ Hybridization

Alcian blue staining was performed as previously described [45,49] with slight modifications. Briefly, embryos were fixed in 4% MEMFA (0.1 M MOPS pH 7.4, 2 mM EGTA, 1 mM MgSO_4_, and 3.7% formaldehyde) and stained in a solution of 0.1% Alcian blue/70% ethanol/0.37% HCl for 1.5 hours. Samples were destained in 70% ethanol/0.37% HCl and cleared in graded series of glycerol solutions. For photography, whole embryos were placed on depression slides. Whole-mount in situ hybridization analyses were conducted as previously described [45] using probes against *foxd3*, *fsta*, and *gata3* [50,51,52]. Briefly, hybridizations were performed overnight at 67 °C with digoxigenin-labeled antisense riboprobes. Following hybridization, the transcript was detected using an anti-digoxigenin antibody conjugated to alkaline phosphatase and chromogenic substrates nitro blue tetrazolium (NBT)/ 5-bromo-4-chloro-3-indolyl phosphate (BCIP).

## 3. Results

### 3.1. Knockdown of Vdrb Impairs Craniofacial Development in Embryos

To determine whether VDR signaling is involved in craniofacial cartilage development, loss-of-function experiments were performed using specific morpholino oligomers (MOs). Subsequently, craniofacial cartilage formation was assessed by Alcian blue staining at 4 dpf (Figure 1). In zebrafish, two *VDR* homologous genes (*vdra* and *vdrb*) are present due to the teleost-specific whole genome duplication [47,53]. In *vdra* MO-injected larvae (morphants), all craniofacial cartilage elements were intact and appeared almost identical to those seen in wild-type controls (Figure 1A,B,E,F). In contrast, *vdrb* morphants displayed deficits in craniofacial structures. Knockdown of *vdrb* resulted in an absence of Meckel’s cartilage and palatoquadrate, a significant reduction of ceratobranchials, and a malformation of ceratohyoid and neurocranium (Figure 1C,G). Weak Alcian blue staining in *vdrb* morphants was further evidence for cartilage hypoplasia. To investigate whether synergistic effects exist between loss-of-*vdra* and -*vdrb*, double knockdown was carried out. Remarkably, coinjection of *vdrb* MO with *vdra* MO produced much more severe defects, which led to the loss of almost all craniofacial cartilage elements (Figure 1D,H). These observations indicate that VDR signaling plays a crucial role in craniofacial cartilage development.

### 3.2. Knockdown of Vdrb Upregulates Fsta Expression in the Pharyngeal Endoderm

The craniofacial cartilages of the zebrafish develop primarily from cranial neural crest cells [24]. To determine whether knockdown of *vdr* affected initial neural crest specification, *foxd3*, the earliest cranial neural crest marker, was examined at 11 hpf. All *vdr* morphants showed normal *foxd3* expression in newly specified premigratory neural crest precursors like wild-type embryos, implying that VDR signaling is dispensable for the early cranial neural crest specification.

BMP signaling is known to play an essential role in differentiation of cranial neural crest cells into the craniofacial cartilages and it has been also suggested that BMP antagonists expressed in the pharyngeal endoderm control BMP activity in craniofacial cartilage precursor cells [35,38,40]. To investigate whether *vdr* knockdown influences BMP signaling, the expression of *fsta*, a BMP antagonist, in the pharyngeal endoderm was examined at 36 hpf by in situ hybridization. Injection of *vdra* MO alone showed no distinguishable effect on the expression of *fsta*, which was similar to those in the control embryos (Figure 2A,B). In contrast, embryos depleted of *vdrb* displayed a dramatic increase of *fsta* expression in the pharyngeal endoderm (Figure 2C). In *vdra*/*b* double morphants, both level and domain of *fsta* expression were comparable to those in *vdrb* single morphants (Figure 2C,D). As a control for in situ hybridization, expression of *gata3*, a BMP-regulated gene in a nonrelevant region was examined (Figure 3). All *vdr* morphants displayed nearly normal expression levels of *gata3* in the posterior part of embryos. Taken together, these results strongly suggest that *vdrb*, not *vdra*, inhibits *fsta* expression in the pharyngeal endoderm to maintain BMP activity during craniofacial cartilage development.

## 4. Discussion

Bone is a classic target of vitamin D and the VDR is present in chondrocytes and osteoblasts [13]. Considerable research has been done focusing on the vitamin D/VDR in bone metabolism. However, relatively little is known regarding the role of VDR signaling in skeletal development during embryogenesis. Being a pilot study, the role of VDR signaling in craniofacial cartilage development was examined in this work. To investigate whether loss-of-*vdr* affects overall gross anatomy of the craniofacial cartilage, Alcian blue staining was applied after the knockdown. For the general evaluation of three-dimensional shapes and sizes of cartilage elements, whole-mounted preparations were more appropriate than flat-mounts in this case. The data presented here clearly demonstrates that VDR signaling is required for craniofacial cartilage development and depletion of *vdrb* leads to severe impairment compared to *vdra* knockdown. It has been reported that knockdown of *vdra*, not *vdrb*, inhibits calcium uptake and leads to delayed ossification of vertebrae [47]. The results of two studies appear to be contradictory, which could be explained by the origin of each skeletal tissue. In zebrafish axial skeleton, craniofacial cartilages mainly develop from cranial neural crest, while vertebrae develop from sclerotome and notochord [54]. It remains possible that two paralogous VDRs may have divergent functions in skeletal development of zebrafish, a teleost [53].

Craniofacial malformations are common structural birth defects and usually caused by perturbations in signaling pathways involved in craniofacial development (reviewed in [55]). BMP signaling pathway plays a crucial role in craniofacial development. It is involved in various processes for craniofacial cartilage development. BMP signaling regulates cranial neural crest cell proliferation, apoptosis, patterning, growth, morphogenesis, and chondrogenic differentiation during craniofacial development [31,34,41]. It is also required for the specification of pharyngeal pouch progenitors which are essential for craniofacial skeleton formation [56]. Defects in BMP signaling are associated with cleft lip/palate in mice and human [57,58]. The present study, for the first time, shows evidence for a crosstalk between VDR and BMP signaling in craniofacial cartilage development: Loss-of-*vdrb* upregulates expression of *fsta*, a BMP antagonist, which may lead to the inhibition of BMP signaling and cause defects in craniofacial cartilages. In zebrafish embryos at 36 hpf, *fsta* is expressed in a broad area including brain and eyes (Figure 2). Since the pharyngeal endoderm (PE) plays an important role in regulating craniofacial development, it would be ideal to quantify changes in PE-specific *fsta* expression using other approach such as qRT-PCR. However, microdissection of PE region for RNA isolation is a bit intricate from the early stage of embryos. In situ hybridization has been routinely used to examine the gene expression level with spatial patterns and to reveal differential regulation of genes following knockdown in zebrafish. In this study, the control and experimental embryos were always processed in parallel and clear overexpression of *fsta* was detected in the PE of *vdrb* morphants, relative to controls under this experimental condition. It has been identified that the transcriptional regulator cascade (Runx3/Egr1/Sox9b) in the PE modulates BMP signaling required for craniofacial cartilage formation by reducing the expression of *fsta* [35]. Intriguingly, the Runx paralogues possess the potential to interact physically and functionally with the VDR [59]. Taken together, this suggests a molecular mechanism by which VDR signaling participates in craniofacial cartilage development: VDR interacts with Runx in the transcriptional cascade which downregulates the expression of *fsta*, thereby maintaining proper BMP signaling activity.

## Figures and Tables

**Figure 1 jdb-07-00013-f001:**
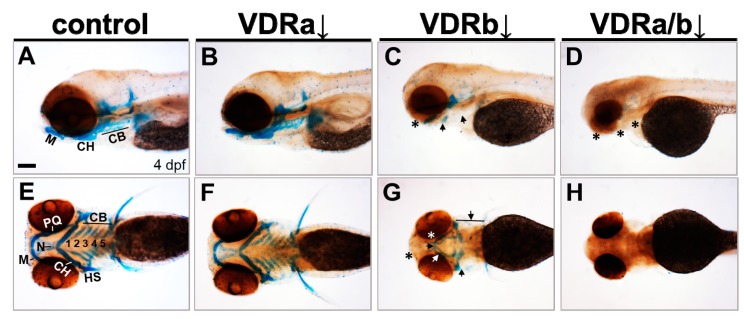
Effects of knockdown of *vdr* on the craniofacial cartilage development. Craniofacial cartilages were stained with Alcian blue in 4 days post-fertilization (dpf) larvae. (**A**,**E**) Wild-type controls. (**B**,**F**) *vdra* morphants. (**C**,**G**) *vdrb* morphants. (**D**,**H**) *vdra*/*b* morphants. Close-up of head lateral (**A**–**D**) and ventral (**E**–**H**) views are shown. Larvae injected with *vdrb* MO display absence, reduction, and malformations of cartilage components compared to controls or *vdra* morphants. These phenotypes are completely penetrant and consistent in *vdrb* morphants. The double knockdown of both *vdra* and *vdrb* leads to more severe defects, complete loss of cartilage. *n* ≥ 10 embryos/condition. Scale bars = 100 μm. Arrows point to reduced or malformed cartilage elements and asterisks denote missing cartilages. M, Meckel’s cartilage; PQ, palatoquadrate; CH, ceratohyal; CB, ceratobranchial; N, neurocranium; HS, hyosymplectic.

**Figure 2 jdb-07-00013-f002:**
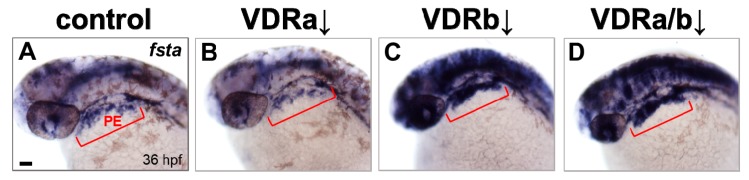
Expression of *fsta* in *vdr* morphants. (**A**) Wild-type control. (**B**) *vdra* morphant. (**C**) *vdrb* morphant. (**D**) *vdra/b* morphant. Expression of *fsta* in pharyngeal arches is upregulated in *vdrb* and *vdra/b* morphants. Upregulation throughout much of the brain, eye, and anterior spinal cord was also noticed in the embryos lacking *vdrb*. Images show dorsolateral views with anterior to the left at 36 hours post-fertilization (hpf). Red brackets indicate the domain of pharyngeal endoderm (PE). Scale bars = 50 μm.

**Figure 3 jdb-07-00013-f003:**
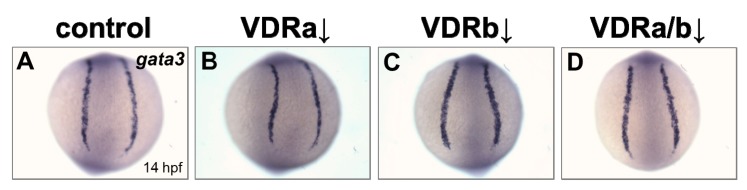
Expression of *gata3* in *vdr* morphants. (**A**) Wild-type control. (**B**) *vdra* morphant. (**C**) *vdrb* morphant. (**D**) *vdra/b* morphant. Images show posterior views at 14 hpf.

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
