# Peer review of "Vitamin D Receptor Signaling Regulates Craniofacial Cartilage Development in Zebrafish"

_jdb, 2019, doi:10.3390/jdb7020013_

Round 1

Reviewer 1 Report

The manuscript is an extension of the work published by this author before. The presented data are very preliminary.

Fig 1 is missing flat mounts, numbers of fish examined and examples pf phenotypoic variations

Fig.2 has no merit since in situ cannot be used to quantify gene expression. Why Q-PCR was not used?

The availability of a genetic model to study VitD role in development is valuable but the presented work is premature to publish.

Author Response

I would like to thank the reviewer for the thoughtful comments and constructive suggestions, which help to improve the quality of this manuscript. Here I provide a point-by-point response to each reviewer’s comments. 

Response to Reviewer 1 comments

General comments. The manuscript is an extension of the work published by this author before. The presented data are very preliminary. …… The availability of a genetic model to study VitD role in development is valuable but the presented work is premature to publish.

Response to the general comments:

The author agrees with the reviewer’s general comments. Yes, it is an extension of the work published previously, but the key points of the current study are not related with the previous works and this study provides new information which was never been published before. As the reviewer pointed out, the availability of a genetic model to study VitD role in development is valuable, especially in the craniofacial development context. Yes, it is preliminary. As shown in the instructions for authors (https://www.mdpi.com/journal/jdb/instructions#submission),

JDB considers several types of publications which include Short communications. It is clearly mentioned “Short Communications of preliminary, but significant, results will be considered.”

Point 1. Fig 1 is missing flat mounts, numbers of fish examined and examples of phenotypic variations.

Response 1:

l  These are not flat-mounted. Images show lateral and ventral views of Alcian blue staining on whole-mount zebrafish larvae. To clarify the issue, the author added the following sentence to the Materials and Methods section (2.3. Alcian blue staining).   

For photography, whole embryos were placed on depression slides.”

l  Numbers of fish examined, and examples of phenotypic variations had been written in Materials and Methods section (2.2.) of the original manuscript. (“To assess phenotypes, at least 10 embryos were examined in each experiment. The phenotypes described in this study were completely penetrant.”) However, as suggested by the reviewer, the information has been added to the figure legend, as shown in the revised manuscript, The sentence now reads:

Figure 1. ………. Larvae injected with vdrb MO display absence, reduction, and malformations of cartilage components compared to controls or vdra morphants. These phenotypes are completely penetrant and consistent in vdrb morphants. The double knockdown of both vdra and vdrb leads to more severe defects, complete loss of cartilage. n ≥ 10 embryos/condition. ….….”

Point 2. Fig 2 has no merit since in situ cannot be used to quantify gene expression. Why Q-PCR was not used?

Response 2:

In zebrafish embryos at 36 hpf, fsta is expressed in the pharyngeal endoderm (PE) region, but it is also expressed in other regions like brain and eyes (Figure 2). If qPCR was performed with the RNA from whole embryos or embryonic heads, the actual data might show a change in expression of fsta within a broad area, but it would be still hard to calculate fold change of fsta expression in pharyngeal arches, essential regions for craniofacial development. As the reviewer commented, in situ may not be the best method to quantify gene expression. However, it has been routinely used to examine the gene expression level with spatial patterns and to reveal differential regulation of genes following overexpression or knockdown. In this study, the control and experimental embryos were always processed in parallel during in situ. A clear overexpression (more intense signal) of fsta was detected in the PE of vdrb morphants, relative to controls (Figure 2). It is fair enough to say upregulation/increase of fsta expression.

Reviewer 2 Report

Please see the attached file for suggestions

Author Response

I would like to thank the reviewer for the thoughtful comments and constructive suggestions, which help to improve the quality of this manuscript. Here I provide a point-by-point response to each reviewer’s comments.

Response to Reviewer 2 comments

Point 1. The materials and methods section is very brief. The morpholino oligomer section, in situ hybridization needs to be described briefly.

Response 1:

As suggested by the reviewer, more information on morpholino oligomers and in situ hybridization has been added to the Materials and Methods section like what is shown below.

l  Gene knockdown experiments using antisense morpholino oligomers (MOs) were carried out as previously described [44,45]. Morpholino oligomers (MOs) were obtained from Gene Tools, LLC. and diluted in Danieau solution (58 mM NaCl, 0.7mM KCl, 0.4 mM MgSO4, 0.6 mM Ca(NO3)2 5.0 mM HEPES, pH 7.6). Zebrafish embryos were injected with 5 ng of each MO at the one-cell stage. All MOs used here have been validated and reported [44,46,47]. To knockdown vdra, a translation blocker (5´-AACGGCACTATTTTCCGTAAGCATC-3´) was used. To knockdown vdrb, a splice blocker (5´-TCCATCACTAGCAGACGAGGGAAGA-3´), which targets the intron2-exon3 (I2E3) junction was used. In all vdrb knockdown experiments, embryos were coinjected with p53 MO to ensure inhibition of non-specific cell death as described [48].

l  Whole-mount in situ hybridization analyses were conducted as previously described [45] using probes against foxd3 [49] and fsta [50]. Briefly, hybridizations were performed overnight at 67°C with digoxigenin-labeled antisense riboprobes. Following hybridization, the transcript was detected using an anti-digoxigenin antibody conjugated to alkaline phosphatase and chromogenic substrates NBT/BCIP.    

Point 2. Are these findings independent of the studies conducted in reference # 44 and 46?

Response 2:

Yes, they are independent. Although all of these are studies on the roles of VDR, each of them deals with totally different aspects. The current study provides new findings which have never been investigated before.

Point 3. Figure 1 needs scale bars. (II. Are E, F, G, H, are same magnification?)

Response 3:

As suggested by the reviewer, a scale bar has been added to Figure 1, as shown in the revised manuscript. (Since A-H are photos with the same magnification, the scale bar has been added to Fig 1A only.)

Point 4. Figure 1. I can see the absence of pectoral fins in figure H. Is there a generalized reduction in cartilage structures in these embryos?

Response 4:

As shown in Figure 1H, one of the most prominent features of the vdra/b double morphants is the abnormal development of pectoral fins. To understand the effects of vdra/b knockdown on the fin development, the expression of several genes involved in limb specification and morphogenesis has been analyzed. vdra/b double morphants showed a severe decrease of expression of fgf24 in fin bud primordia at 22 hpf (unpublished data). This data suggests VDR signaling may play a role in the very early stage of pectoral fin development. The absence of pectoral fins in vdra/b double morphants is possibly caused by defects in the initial specification of the fin bud primordia. In addition, the pectoral fin endoskeleton arises in the fin bud mesenchyme, while craniofacial cartilages mainly develop from the cranial neural crest. The molecular mechanism by which VDR signaling participates in craniofacial cartilage development could be distinct from the mechanism which controls pectoral fin development.

Point 5. Figure 1. Use asterisks or arrows to highlight the defects.

Response 5:

As suggested by the reviewer, the author has added asterisk and arrows to Figure 1, as shown in the revised manuscript. The following sentence has been added to the figure legend as well.

Arrows point to reduced or malformed cartilage elements and asterisks denote missing cartilages.”

Point 6. The discussion section needs to be discussed on the role of Bmp pathway signaling on cartilage development.

Response 6:

As suggested by the reviewer, the role of Bmp pathway signaling on cartilage development has been discussed more. The following has been added to the Discussion section.

“BMP signaling pathway plays a crucial role in craniofacial development. It is involved in various processes for craniofacial cartilage development. BMP signaling regulates cranial neural crest cell proliferation, apoptosis, patterning, growth, morphogenesis, and chondrogenic differentiation during craniofacial development [31,34,41]. It is also required for the specification of pharyngeal pouch progenitors which are essential for craniofacial skeleton formation [56]”

Point 7. Line 47-49 Rewrite the sentence for clarity.

Response 7:

The author has rewritten the sentence, as shown in the revised manuscript (Line 46-49). The sentence now reads:

“The formation of craniofacial structures is controlled by multiple signaling molecules including FGF, SHH, WNT, Endothelin-1 (Edn1), Notch, and BMP [26-30]. BMP signaling has long been recognized as a key regulator of craniofacial development [31,32].”

Point 8. Line 104-106 Rewrite the sentence for clarity.

Response 8:

The author has rewritten the sentence, as shown in the revised manuscript (Line 114-115). The sentence now reads:

To determine whether knockdown of vdr affected initial neural crest specification, foxd3, the earliest cranial neural crest marker, was examined at 11 hpf.”

Round 2

Reviewer 1 Report

I have reviewed the second version of the manuscript. I am not satisfied with the response of the author

- I maintain that the work, why novel, is too poorly documented at this time and too preliminary to merit a publication

-I know what the flat mounts are and I asked the author to provide such. Currently the Fig 1 is not informative

- there is NO WAY that WISH can be used to quantify anything besides saying "apparently less" or "apparently more". Just like immunofluorescence cannot be used as a quantification tool. The in situs suggest an overall increase in fsta and vdr expression so the Q-PCR will give adequate results.

While I understand that sometimes it is imperative to publish preliminary data, one needs to separate that from a "salami" science. There is really no need for that here

Author Response

I would like to thank you for your advice on my manuscript.

Here are the major points of reviewer 1’s comments (Round 2) and my responses.

Point 1 – “I know what the flat mounts are and I asked the author to provide such. Currently the Fig 1 is not informative.”

Response 1: Conventionally, most of zebrafish embryo craniofacial cartilage papers present Alcian blue staining data in whole-mounted preparations. If anyone simply conducts a PubMed search using “zebrafish” AND “craniofacial” AND “Alcian blue” as query, the result displays 10 items currently (6-Jun-19). Among these only 1 paper (out of 10) contains the flat mount. In other words, the other 9 papers including Zhang et al. (2019) and Staal et al. (2018) published in high profile journals provide only whole mounts. Of course, flat mounts allow the specimen to be photographed in one focal plane, producing very informative depictions of each cartilage. However, flat mounting disturbs three-dimensional relationships (Westerfield et al., 2009). In my study, the purpose of Alcian blue staining was to investigate general effects of knockdown of vdr on the craniofacial cartilage development. Therefore, overall gross anatomy of the craniofacial cartilage in 3D was more important. The reviewer says “not informative”; on the contrary, Fig.1 is very clear and speaks for itself in my opinion. Anyone who sees and compares the images in Fig.1 can easily catch the message - knockdown of vdrb leads to abnormal development (absence, reduction, and malformations) of cartilage components.

Point 2 – “There is NO WAY that WISH can be used to quantify anything besides saying "apparently less" or "apparently more". Just like immunofluorescence cannot be used as a quantification tool. The in situs suggest an overall increase in fsta and vdr expression so the Q-PCR will give adequate results.”

Response 2: Again, I agree. The whole mount in situ hybridization (WISH) is not the best tool to quantify gene expression. But, as the reviewer commented, at least it says more/less expression or suggests increase/decrease in gene-of-interest expression. In fact, the way that I described WISH results in the manuscript were: “elevated expression of fsta”, “a dramatic increase of fsta expression in the pharyngeal endoderm”, and “Expression of fsta in pharyngeal arches is upregulated.” Actually, there is important reason why I think WISH is more appropriate tool than Q-PCR in this case. Since the pharyngeal endoderm (PE) plays an important role in regulating craniofacial development, the specific aim is to determine whether vdr knockdown affects the expression of fsta in the PE. The tricky thing is fsta is also expressed in other regions like brain and eyes and it is not easy to isolate PE region only at 36 hpf. Q-PCR using RNA from a broad area (e.g., embryonic head) might not give the right answer for the specific question.